# Simple Augmentation Goes a Long Way: ADRL for DNN Quantization

**Lin Ning**[*][†]
North Carolina State University
Raleigh, NC, USA
lning@ncsu.edu

**Guoyang Chen**
Alibaba Group
Sunnyvale, CA, USA
g.chen@alibaba-inc.com

**Weifeng Zhang**
Alibaba Group
Sunnyvale, CA, USA
weifeng.z@alibaba-inc.com

**Xipeng Shen**
North Carolina State University
Raleigh, NC, USA
xshen5@ncsu.edu

## Abstract

Mixed precision quantization improves DNN performance by assigning different layers with different bit-width values. Searching for the optimal bit-width for each layer, however, remains a challenge. Deep Reinforcement Learning (DRL) shows some recent promise. It however suffers instability due to function approximation errors, causing large variances in the early training stages, slow convergence, and suboptimal policies in the mixed precision quantization problem. This paper proposes augmented DRL (ADRL) as a way to alleviate these issues. This new strategy augments the neural networks in DRL with a complementary scheme to boost the performance of learning. The paper examines the effectiveness of ADRL both analytically and empirically, showing that it can produce more accurate quantized models than the state of the art DRL-based quantization while improving the learning speed by 4.5-64$\times$.

## 1 Introduction

By reducing the number of bits needed to represent a model parameter of Deep Neural Networks (DNN), quantization (Lin et al., 2016; Park et al., 2017; Han et al., 2015; Zhou et al., 2018; Zhu et al., 2016; Hwang & Sung, 2014; Wu et al., 2016; Zhang et al., 2018; Köster et al., 2017; Ullrich et al., 2017; Hou & Kwok, 2018; Jacob et al., 2018) is an important way to reduce the size and improve the energy efficiency and speed of DNN. Mixed precision quantization selects a proper bit-width for each layer of a DNN, offering more flexibility than fixed precision quantization.

A major challenge to mixed precision quantization (Micikevicius et al., 2017; Cheng et al., 2018) is the configuration search problem, that is, how to find the appropriate bit-width for each DNN layer efficiently. The search space grows exponentially as the number of layers increases, and assessing each candidate configuration requires a long time of training and evaluation of the DNN.

Research efforts have been drawn to mitigate the issue for help better tap into the power of mixed precision quantization. Prior methods mainly fall into two categories: (i) automatic methods, such as reinforcement learning (RL) (Lou et al., 2019; Gong et al., 2019; Wang et al., 2018; Yazdanbakhsh et al., 2018; Cai et al., 2020) and neural architecture search (NAS) (Wu et al., 2018; Li et al., 2020), to learn from feedback signals and automatically determine the quantization configurations; (ii) heuristic methods to reduce the search space under the guidance of metrics such as weight loss or Hessian spectrum (Dong et al., 2019; Wu et al., 2018; Zhou et al., 2018; Park et al., 2017) of each layer.

Comparing to the heuristic method, automatic methods, especially Deep Reinforcement Learning (DRL), require little human effort and give the state-of-the-art performance (e.g., via actor-critic set-

---

[*]This work was done when Lin Ning was an intern at Alibaba Group.
[†]Now at Google Research.

ting (Whiteson et al., 2011; Zhang et al., 2016; Henderson et al., 2018; Wang et al., 2018)). It however suffers from overestimation bias, high variance of the estimated value, and hence slow convergence and suboptimal results. The problem is fundamentally due to the poor function approximations given out by the DRL agent, especially during the early stage of the DRL learning process (Thrun & Schwartz, 1993; Anschel et al., 2017; Fujimoto et al., 2018) when the neural networks used in the DRL are of low quality. The issue prevents DRL from serving as a scalable solution to DNN quantization as DNN becomes deeper and more complex.

This paper reports that simple augmentations can bring some surprising improvements to DRL for DNN quantization. We introduce *augmented DRL* (ADRL) as a principled way to significantly magnify the potential of DRL for DNN quantization. The principle of ADRL is to augment the neural networks in DRL with a complementary scheme (called *augmentation scheme*) to complement the weakness of DRL policy approximator. Analytically, we prove the effects of such a method in reducing the variance and improving the convergence rates of DRL. Empirically, we exemplify ADRL with two example augmentation schemes and test on four popular DNNs. Comparisons with four prior DRL methods show that ADRL can shorten the quantization process by 4.5-64× while improving the model accuracy substantially. It is worth mentioning that there is some prior work trying to increase the scalability of DRL. Dulac-Arnold et al. (2016), for instance, addresses large discrete action spaces by embedding them into continuous spaces and leverages nearest neighbor to find closest actions. Our focus is different, aiming to enhance the learning speed of DRL by augmenting the weak policy approximator with complementary schemes.

## 2  BACKGROUND

**Deep Deterministic Policy Gradient (DDPG)**  A standard reinforcement learning system consists of an agent interacting with an environment $\mathcal{E}$. At each time step $t$, the agent receives an observation $x_t$, takes an action $a_t$ and then receives an award $r_t$. Modeled with Markov decision process (MDP) with a state space $\mathcal{S}$ and an action space $\mathcal{A}$, an agent's behavior is defined by a policy $\pi : \mathcal{S} \rightarrow \mathcal{A}$. A state is defined as a sequence of actions and observations $s_t = (x_1, a_2, \cdots, a_{t-1}, x_t)$ when the environment is (partially) observed. For DNN quantization, the environment is assumed to be fully observable ($s_t = x_t$). The return from a state $s$ at time $t$ is defined as the future discounted return $R_t = \sum_{i=t}^{T} \gamma^{i-t} r(s_i, a_i)$ with a discount factor $\gamma$. The goal of the agent is to learn a policy that maximizes the expected return from the start state $\mathcal{J}(\pi) = \mathbb{E}[\mathcal{R}_1|\pi]$ . An RL agent in continuous action spaces can be trained through the actor-critic algorithm and the deep deterministic policy gradient (DDPG). The parameterized actor function $\mu(s|\theta^\mu)$ specifies the current policy and deterministically maps a state $s$ to an action $a$. The critic network $Q(s,a)$ is a neural network function for estimating the action-value $\mathbb{E}[R_t|s_t = s, a_t = a, \pi]$; it is parameterized with $\theta^Q$ and is learned using the Bellman equation as Q-learning. The critic is updated by minimizing the loss

$$L(\theta^Q) = \mathbb{E}[(y_t - Q(s_t, a_t|\theta^Q))^2], \quad where \quad y_t = r(s_t, a_t) + \gamma Q(s_{t+1}, \mu(s_{t+1}|\theta^\mu)|\theta^Q). \quad (1)$$

The actor is updated by applying the chain rule to the expected return $\mathcal{J}$ with respect to its parameters:

$$\nabla_{\theta^\mu} \mathcal{J} \approx \mathbb{E}[\nabla_a Q(s, a|\theta^Q)|_{s=s_t, a=\mu(s_t)} \nabla_{\theta_\mu} \mu(s|\theta^\mu)|_{s=s_t}]. \quad (2)$$

**DDPG for Mixed Precision Quantization**  To apply the DRL to mixed precision quantization, previous work, represented by HAQ (Wang et al., 2018), uses DDPG as the agent learning policy. The environment is assumed to be fully observed so that $s_t = x_t$, where the observation $x_t$ is defined as $x_t = (l, c_{in}, c_{out}, s_{kernel}, s_{stride}, s_{feat}, n_{params}, i_{dw}, i_{w/a}, a_{t-1})$ for convolution layers and $x_t = (l, h_{in}, h_{out}, 1, 0, s_{feat}, n_{params}, 0, i_{w/a}, a_{t-1})$ for fully connected layers. Here, $l$ denotes the layer index, $c_{in}$ and $c_{out}$ are the numbers of input and output channels for the convolution layer, $s_{kernel}$ and $s_{stride}$ are the kernel size and stride for the convolution layer, $h_{in}$ and $h_{out}$ are the numbers of input and output hidden units for the fully connected layer, $n_{params}$ is the number of parameters, $i_{dw}$ and $i_{w/a}$ are binary indicators for depth-wise convolution and weight/activation, and $a_{t-1}$ is the action given by the agent from the previous step.

In time step $t-1$, the agent gives an action $a_{t-1}$ for layer $l-1$, leading to an observation $x_t$. Then the agent gives the action $a_t$ for layer $l$ at time step $t$ given $x_t$. The agent updates the actor and critic networks after one episode following DDPG, which is a full pass of all the layers in the target neural network for quantization. The time step $t$ and layer index $l$ are interchangeable in this scenario.

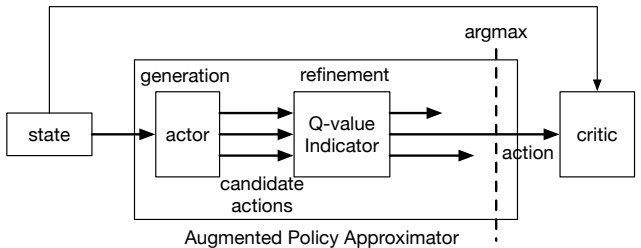

Figure 1: Illustration of the augmented policy approximator

The systems use a continuous action space for precision prediction. At each time step $t$, the continuous action $a_t$ is mapped to the discrete bit value $b_k$ for layer $k$ using $b_k = round(b_{min} - 0.5 + a_k \times (b_{max} - b_{min} + 1))$. The reward function is computed using $\mathcal{R} = \lambda \times (acc_{quant} - acc_{origin})$.

## 3    AUGMENTED DEEP REINFORCEMENT LEARNING (ADRL)

This section explains our proposal, ADRL. It builds upon the default actor-critic algorithm and is trained with DDPG. In the original actor-critic algorithm, the policy approximator is the actor networks, which generates one action and feeds it into the critic networks. The essence of ADRL is to augment the policy approximator with a supplementary scheme. The scheme may be in an arbitrary form, constructed with domain knowledge or other mechanisms. It should boost the weak policy approximator in accuracy especially when it is not yet well trained, and hence help efficiently reduce the variance and accelerate the convergence of the DRL algorithm.

### 3.1    DESIGN

**Components of ADRL**   Figure 1 illustrates the general flow of ADRL that employs post augmentation. The augmented policy approximator consists of two main components: an expanded actor network and a refinement function. The expansion of the actor network makes it output multiple rather than one candidate action; the refinement function derives the most promising candidate by processing those actions and feeds it to the critic networks. The two components can be formally described as follows.

*Actions generation* The expanded actor function, $\hat{\mu}_{\theta\hat{\mu}} : \mathcal{S} \to \mathbb{R}^{k \times n}$, maps a state from the state space $\mathbb{R}^m$ to $k$ ($k > 1$) actions in a continuous actions space $\mathbb{R}^n$: $A_k = \hat{\mu}(\mathbf{s}|\theta^{\hat{\mu}})$ where $A_k = [\mathbf{a}_1^T, \cdots, \mathbf{a}_k^T]^T$. The expansion can be done by modifying the last layer of the actor network (Sec 3.3). The outputs of this actor function serve as the candidate actions to the refinement component.

*Action refinement.* The action refinement function, $g : \mathbb{R}^{k \times n} \to \mathbb{R}^n$, derives a promising action from the candidate actions. A simple form of derivation is selection, that is, $\mathbf{a}^* = \arg\max_{\mathbf{a}=A_{k,i}} Q(\mathbf{s}, \mathbf{a})$ Depending on how well the critic network is trained, the critic may not be able to consistently give a good estimation of $Q(\mathbf{s}, \mathbf{a})$, especially at the early training stage. The augmented policy may use a Q-value indicator $\widetilde{Q}_{\mathcal{E}}(\mathbf{a})$ whose output depends on only the action and the property of the environment. Hence, $g(A_k) = \arg\max_{\mathbf{a}=A_{k,i}} \widetilde{Q}_{\mathcal{E}}(\mathbf{a})$ The choice of $\widetilde{Q}_{\mathcal{E}}(\mathbf{a})$ also depends on the specific tasks the ADRL is trying to solve.

Combining the generation function and the refinement function, the augmented policy estimator can be represented as $\pi_{\theta\hat{\mu}}(\mathbf{s}) = g \circ \hat{\mu}_{\theta\hat{\mu}}$.

**Training with DDPG**   We train parameters for the actor and the critic networks using DDPG. Although the augmented policy $\pi_{\theta\hat{\mu}}$ consists of an actor network $\hat{\mu}_{\theta\hat{\mu}}$ and a refinement function $g$, the training of the full policy follows the policy gradient of $\hat{\mu}_{\theta\hat{\mu}}$ because the effects of $g$ are deterministic aspects of the environment $\mathcal{E}$ (Dulac-Arnold et al., 2015).

The actor and the critic network are trained with variations of formulae (1) and (2):

$$L(\theta^Q) = \mathbb{E}[(y_t - Q(s_t, \hat{\mu}(s_t|\theta^{\hat{\mu}})|\theta^Q))^2], \ where \ y_t = r(s_t, \pi_{\theta\hat{\mu}}(s_t)) + \gamma Q(s_{t+1}, \hat{\mu}(s_{t+1}|\theta^{\hat{\mu}})|\theta^Q)$$

---

**Algorithm 1** Augmented Policy

---

1: State $\mathbf{s}$ previously received from the environment $\mathcal{E}$
2: $A_k = \mu(s|\theta^\mu)$ (generating $k$ candidate actors)
3: $\mathbf{a} = g(A_k)(\mathbf{a})$ (refines the choice of the action with $g(A_k) = \arg\max_{\mathbf{a}=A_{k,i}} \widetilde{Q}_\mathcal{E}(\mathbf{a})$)
4: Apply $\mathbf{a}$ to environment; receive $r, s'$

---

$$\nabla_{\theta^{\hat{\mu}}} \mathcal{J} \approx \mathbb{E}[\nabla_{A_k} Q(s, A_k|\theta^Q)|_{s=s_t, A_k=\hat{\mu}(s_t)} \cdot \nabla_{\theta_{\hat{\mu}}} \hat{\mu}(s|\theta^{\hat{\mu}})|_{s=s_t}]. \tag{3}$$

## 3.2 Effects on Variance and Learning Speed

**Variance Reduction** The actor-critic algorithm for mixed-precision quantization is a combination of Q-learning and function approximation. Therefore, it suffers from various types of function approximation errors which affect the training stability. Analysis below follows the terminology from previous work (Thrun & Schwartz, 1993).

**Definition 1** *If actions derived by the augmented policy approximator leads to higher rewards comparing to actions given by the default actor network, the augmentation is an* effective augmentation.

**Definition 2** *Let $Q(s_t, a_t|\theta^Q)$ be the value function of DRL at time step $t$, $Q^*(s_t, a_t) = \max_\pi Q^\pi(s_t, a_t)$ be the optimal value function, and $y_{s,a}^t = r(s_t, a_t) + \gamma Q(s_{t+1}, a_{t+1}|\theta^Q)$ be the DRL target. Then the function approximation error denoted as $\delta_t$ can be decomposed as $\Delta_t = Q(s_t, a_t|\theta^Q) - Q^*(s_t, a_t) = Q(s_t, a_t|\theta^Q) - y_{s,a}^t + y_{s,a}^t - Q^*(s_t, a_t)$ The first two items form the Target Approximation Error (TAE), denoted as $Z_{s,a}^t = Q(s_t, a_t|\theta^Q) - y_{s,a}^t$.*

**Proposition 1** *Assuming that TAE is a random process (Thrun & Schwartz, 1993) such that $E[Z_{s,a}^t] = 0$, $Var[Z_{s,a}^t] = \sigma_s^2$, and $Cov[Z_{s,a}^{t_1}, Z_{s',a'}^{t_2}] = 0$ for $t_1 \neq t_2$, then an* effective augmentation *leads to a smaller variance of the estimated Q value.*

*Proof:* The estimated Q value function can be expressed as $Q(s_t, a_t|\theta^Q) = Z_{s,a}^t + r(s_t, a_t) + \gamma Q(s_{t+1}, a_{t+1}|\theta^Q) = Z_{s,a}^t + r(s_t, a_t) + \gamma[Z_{s,a}^{t+1} + r(s_{t+1}, a_{t+1}) + \gamma Q(s_{t+2}, a_{t+2}|\theta^Q)] = \sum_{i=t}^T \gamma^{i-t}[r(s_i, a_i) + Z_{s,a}^i]$. It is a function of rewards and TAEs. Therefore, the variance of the estimated Q value will be proportional to the variance of the reward and TAE, such that $Var[Q(s_t, a_t|\theta^Q)] = \sum_{i=t}^T \gamma^{i-t}(Var[r(s_i, a_i)] + Var[Z_{s,a}^i])$. Let $[r_{min}, r_{max}]$ and $[\widetilde{r}_{min}, \widetilde{r}_{max}]$ be the range of rewards associated with the default actor network and the augmented policy approximator respectively. With an *effective augmentation*, we have $r_{min} < \widetilde{r}_{min}$, and $r_{max} = \widetilde{r}_{max}$. It is easy to derive that $Var[\widetilde{r}] < Var[r]$. Since $Var[Z_{s,a}^i]$ is the same for both of the cases, an *effective augmentation* leads to a smaller variance of the estimated Q value due to smaller reward variance. $\square$

In ADRL, we use a Q-value indicator to help choose actions that tend to give higher quantization accuracy, leading to a higher reward. Therefore, it helps reduce the variance of the estimated Q value, as confirmed by the experiment results in Section 4.

**Effects on Learning Speed**

**Proposition 2** *An* effective augmentation *leads to larger step size and hence faster updates of the critic network.*

*Proof:* The critic network is trained by minimizing loss as shown in Eq. 1. Let $\delta_t = y_t - Q(s_t, a_t|\theta^Q)$. By differentiating the loss function with respect to $\theta^Q$, we get the formula for updating the critic network as $\theta_{t+1}^Q = \theta_t^Q + \alpha_{\theta^Q} \cdot \delta_t \cdot \nabla_{\theta^Q} L(\theta^Q)$. Let $\pi_{\theta^{\hat{\mu}}}$ be the augmented policy approximator and $\mu_{\theta^\mu}$ be the default actor network. With an *effective augmentation*, the augmented policy approximator should generate an action that leads to higher rewards such that $r(s_t, \pi_{\theta^{\hat{\mu}}}(s_t)) >= r(s_t, \mu_{\theta^\mu}(s_t))$, and has a larger $Q(s_{t+1}, a_{t+1})$. Assuming the same starting point for both $\pi_{\theta^{\hat{\mu}}}$ and $\mu_{\theta^\mu}$, which means that $Q(s_t, a_t)$ are the same for both of the two policies, we get $\delta_t^{\pi_{\theta^{\hat{\mu}}}} > \delta_t^{\mu_{\theta^\mu}}$. Therefore, $\pi_{\theta^{\hat{\mu}}}$ leads to larger $\alpha_{\theta^Q} \cdot \delta_t \cdot \nabla_{\theta^Q} L(\theta^Q)$, and thus a larger step size and hence faster updates to $\theta^Q$, the parameters in the critic network. $\square$

Meanwhile, an *effective augmentation* helps DRL choose better-performing actions especially during the initial phases of the learning. Together with the faster updates, it tends to make the learning converge faster, as consistently observed in our experiments reported in Section 4.

### 3.3 Application on Mixed-Precision Quantization

We now present how ADRL applies to mixed precision quantization. The key question is how to design the augmented policy approximator for this specific task. Our design takes the HAQ (Wang et al., 2018) as the base. For the actor networks, our expansion is to the last layer. Instead of having only one neuron for the last layer, our action generator has $k$ neurons corresponding to $k$ candidate actions. We give a detailed explanation next.

**Q-value Indicator**   The augmentation centers around a Q-value indicator that we introduce to help select the most promising action from the candidates generated by the expanded actor function.

To fulfill the requirement of the ADRL, the Q-value indicator should be independent from the learnable parameters of the actor and critic networks. Its output shall be determined by the action $\mathbf{a}$ and the environment $\mathcal{E}$. Since the Q-value is the expected return, it is directly connected with the inference accuracy after quantization ($acc_{quant}$). In this work, we experiment with two forms of Q-value indicators. They both estimate the relative fitness of the candidate quantization levels in terms of the influence on $acc_{quant}$. Their simplicities make them promising for actual adoptions.

*Profiling-based indicator.* The first form is to use the inference accuracy after quantization ($acc_{quant}$) such that $Q_{\mathcal{E}}(\mathbf{a}) \sim acc_{quant|quantize(W_l, b(a_l))}$. At each time step $t$, the action generator generates $k$ action candidates for layer $l$, where $l = t$. For each action $\mathbf{a}_i = A_{k,i}$, the Q-value indicator computes the corresponding bit value $b_i$ and quantizes layer $l$ to $b_i$ bits while keeping all other layers unchanged. The quantized network is evaluated using a test dataset and the resulting accuracy is used as the indication of the fitness of the quantization level. The candidate giving the highest accuracy is selected by the action refinement function $g$. When there is a tie, the one that leads to a smaller network is selected by $g$.

*Distance-based indicator.* The second form is the distance between the original weight and the quantized weight given the bit value $b(a_l)$ for layer $l$. We explore two forms of distance: L2 norm ($Q_{\mathcal{E}}(\mathbf{a}) \sim ||W_l - quantize(W_l, b(a_l))||$) and KL-divergence (Wikipedia contributors, 2020) ($Q_{\mathcal{E}}(\mathbf{a}) \sim \mathcal{D}_{KL}(W_l || quantize(W_l, b(a_l)))$). Both of them characterize the distance between the two weight distributions, and hence how sensitive a certain layer is with respect to a quantization level. As the distance depends only on the quantized bit value $b_l$ and the original weight, all the computations can be done offline ahead of time. During the search, the refinement function $g$ selects the action that gives the smallest KL-divergence for the corresponding layer from the candidate pool.

**Implementation Details**   The entire process of mixed-precision quantization has three main stages: searching, finetuning, and evaluation. The main focus of this work is the search stage, where we run the ADRL for several episodes to find a quantization configuration that satisfies our requirements (e.g., weight compression ratio). Then we finetune the model with the optimal configuration for a number of epochs until the model converges. A final evaluation step gives the accuracy of the quantized model achieves.

In each episode of the search stage, the agent generates a quantization configuration with the guidance of both the quantized model accuracy and the model compression ratio. First, the Q-value indicator selects the candidate network that gives the highest accuracy for each layer. After selecting the bit values for all the layers, the agent checks whether the quantized network fulfills the compression ratio requirement. If not, it will decrease the bit value layer by layer starting from the one that gives the least accuracy loss to generate a quantized model that fulfills the compression ratio requirement. The accuracy of this quantized model is used to compute the reward and to train the agent.

*Memoization.* When using the profiling based indicator, the agent needs to run inference on a subset of test dataset for each candidate action. To save time, we introduce memoization into the process. We use a dictionary to store the inference accuracy given by different bit-width values for each layer. Given a candidate action $a_i$ for layer $l$, the agent maps it to a discrete bit value $b_i$ and checks if the inference accuracy of quantizing weights of layer $l$ to $b_i$ bits is stored in the corresponding dictionary. If so, the agent simply uses the stored value as the indicated Q-value for action $a_i$. Otherwise, it

quantizes layer $l$ to $b_i$ bits, evaluates the network to get the accuracy as the indicated Q-value, and adds the $b_i : acc_{quant}(b_i)$ pair into the dictionary.

*Early termination.* A typical design for the searching stage runs the RL agent for a fixed number of episodes and outputs the configuration that gives the highest reward. We observed that in the process of searching for optimal configurations, the time it takes for the $acc_{quant}$ to reach a stable status could be much less than the pre-set number of episodes. Especially when using ADRL, the agent generates actions with high rewards at a very early stage. Therefore, we introduce an early termination strategy, which saves much computation time spending on the search process.

The general idea for early termination is to stop the search process when the inference accuracy has small variance among consecutive runs. We use the standard derivation among $N$ runs to measure the variance. If it is less than a threshold (detailed in Sec 4), the search process terminates.

*Time complexity.* Let $T_{gen}$ be the time of generating actions for each layer and $T_{eval}$ be the time spending on evaluating the quantized network once. The time complexities of the original DRL method and ADRL without memorization are $N \cdot (L \cdot T_{gen} + T_{eval})$ and $N' \cdot (L \cdot T_{gen} + (kL+1) \cdot T_{eval})$, where $N$ and $N'$ are the number of search episodes for the original DRL method and ADRL, $k$ is the number of candidate actions in ADRL for each layer and $L$ is the number of layers of the target network. The overhead brought by the Q-value indicator is $N'kL \cdot T_{eval}$ since it needs to evaluate the network accuracy for each candidate action in each layer. With *memorization*, assuming $n$ is the number of possible bit values in the search space, this overhead is largely reduced as the Q-value indicator only needs to evaluate the network $n$ times for each layer during the entire search stage. Therefore, if the network needs to be quantized to at most 8 bits, the overhead becomes $8L \cdot T_{eval}$. The time complexity is hence reduced to $N' \cdot (L \cdot T_{gen} + T_{eval}) + 8L \cdot T_{eval}$. With early termination, $N'$ is much smaller than $N$. Therefore, the total time spending on the search stage with ADRL is much smaller than that of the original DRL based method, which is demonstrated in the experiment results as shown in Sec 4.

## 4 EXPERIMENTAL RESULTS

To evaluate the efficiency of *augmented DRL* on mixed precision quantization of DNNs, we experiment with four different networks: CifarNet (Krizhevsky, 2012), ResNet20 (He et al., 2016), AlexNet (Krizhevsky et al., 2012) and ResNet50 (He et al., 2016). The first two networks work on Cifar10 (Krizhevsky, 2012) dataset while the last two work on the ImageNet (Deng et al., 2009) dataset. The experiments are done on a server with an Intel(R) Xeon(R) Platinum 8168 Processor, 32GB memory and 4 NVIDIA Tesla V100 32GB GPUs.

The entire process of DRL based mixed-precision quantization consists of three main stages: search, finetuning, and evaluation. The search stage has $E$ episodes. Each episode consists of $T$ time steps, where $T$ equals the number of layers of the neural network. At each time step $t$, the RL algorithm selects a bit-width value for layer $l = t$. After finishing 1 episode, it generates a mixed-precision configuration with the bit-width values for all the layers, quantizes the network accordingly, and evaluates the quantized network to calculate the reward. It then updates the actor and critic parameters and resets the target DNN network to the original non-quantized one. The finetuning stage follows the search stage. In this stage, we quantize the DNN network using the mixed-precision configuration selected by the RL algorithm at the end of the search stage and finetune the network for $N$ epochs. The final stage is evaluation, which runs the finetuned quantized model on the test dataset. The resulting inference accuracy is the metric for the quality of the quantized network.

We compare our ADRL with four prior mixed-precision quantization studies. Besides the already introduced HAQ (Wang et al., 2018), the prior studies include ReLeQ (Yazdanbakhsh et al., 2018), HAWQ (Dong et al., 2019), and ZeroQ (Cai et al., 2020). ReLeQ (Yazdanbakhsh et al., 2018) uses a unique reward formulation and shaping in its RL algorithm so that it can simultaneously optimize for two objectives (accuracy and reduced computation with lower bit width) with a unified RL process. HAWQ (Dong et al., 2019) automatically selects the relative quantization precision of each layer based on the layer's Hessian spectrum and determines the fine-tuning order for quantization layers based on second-order information. ZeroQ (Cai et al., 2020) enables mix-precision quantization without any access to the training or validation data. It uses a Pareto frontier based method to automatically determine the mixed-precision bit setting for all layers with no manual search involved.

Our experiments use layer-wise quantization, but we note that ADRL can potentially augment policy approximators in quantizations at other levels (e.g. kernel-wise Lou et al. (2019)) in a similar way.

For ADRL, we experimented with both P-ADRL (profiling-based) and D-ADRL (distance-based). P-ADRL outperforms D-ADRL. We hence focus the overall results on P-ADRL, but offer the results of D-ADRL in the detailed analysis.

As ADRL is implemented based on HAQ, we collect detailed results on all experimented networks on HAQ (code downloaded from the original work (Wang et al., 2018)) to give a head-to-head comparison. For the other three methods, we cite the numbers reported in the published papers. The default setting of HAQ is searching for 600 episodes and finetuning for 100 epochs. We also apply early termination to HAQ (HAQ+E), where, the termination criterion is that the coefficient of variance of the averaged inference accuracy over 10 episodes is smaller than 0.01 for at least two consecutive 10-episode windows. In all experiments, $k$ (number of candidate actions) is 3 for ADRL. We report the quality of the quantization first, and then focus on the speed and variance of the RL learning.

Table 1: Comparison of quantization results. (Comp: compression ratio; Acc $\Delta$: accuracy loss; '-': no results reported in the original paper)

|  | ResNet20 | | AlexNet | | ResNet50 | |
| --- | --- | --- | --- | --- | --- | --- |
|  | Comp | Acc $\Delta$ | Comp | Acc $\Delta$ | Comp | Acc $\Delta$ |
| **ADRL (i.e., P-ADRL)** | 10.3X | 0 | 10.1X | 0 | 10.0X | 0 |
| HAQ (Wang et al., 2018) | 10.1X | 0.41 | 10.8X | 0 | 10.1X | 1.168 |
| ReLeQ (Yazdanbakhsh et al., 2018) | 1.88X | 0.12 | 3.56X | 0.08 | - | - |
| HAWQ (Dong et al., 2019) | 13.1X | 0.15 | - | - | 12.8X | 1.91 |
| ZeroQ (Cai et al., 2020) | - | - | - | - | 8.4X | 1.64 |

**Compression Ratios and Accuracy**     Table 1 compares the compression ratios and accuracy. For the effects of the augmentation of the approximator, ADRL is the only method that delivers a comparable compression ratio without causing any accuracy loss on all three networks. Note that ADRL is more as a complement rather than a competitor to these work: The augmentation strategy it introduces can help complement the approximators in the existing DRL especially when they are not yet well trained.

**Learning Speed**     Table 2 reports the learning speed in each of the two major stages (search, finetune) and the overall learning process. The reported time is the total time before convergence. For the search stage, the time includes the overhead brought by the Q-value indicator. As shown in column 5 and 8, the search stage is accelerated by 6–66$\times$, while the finetune stage by up to 20$\times$. CifarNet is an exception. For its simplicity, all methods converge in only one epoch in finetuning; because HAQ has a larger compression ratio as shown in Figure 2, neither P-ADRL nor HAQ+E has a speedup in finetuning over HAQ. But they take much fewer episodes to search and hence still get large overall speedups. Overall, P-ADRL shortens RDL learning by HAQ by a factor of 4.5 to 64.1, while HAQ+E gets only 1.2 to 8.8.

The reasons for the speedups come from the much reduced numbers of episodes needed for the DRL to converge in search, and the higher accuracy (and hence earlier termination) that P-ADRL gives in the finetuning process. We next give a detailed analysis by looking into each of the two stages in further depth.

**Search Stage**     Figure 2 shows the inference accuracy (acc), the coefficient of variance (CV) of the inference accuracy, and the compression ratio in terms of the weight ratio (wratio) of the network for the search stage. The first row of Figure 2 gives the inference accuracy results in the search stage. we can see that the configurations found by P-ADRL give comparable inference accuracy for CifarNet and ResNet20 on Cifar10 dataset and better accuracy for AlexNet and ResNet50 on ImageNet dataset comparing to that of HAQ. Also, with early termination, our strategy terminates with a much smaller number of search episodes comparing to HAQ. As shown in column 3 of Table 2, the numbers of search episodes of P-ADRL are 19.1% - 34.5% of that of HAQ+E and 2.2% - 12.3% of that of the original HAQ. The performance of D-ADRL is between that of P-ADRL and HAQ on CifarNet, ResNet20 and ResNet50 and the worst on AlexNet. P-ADRL is more promise than D-ADRL does.

Table 2: Execution time of each stage of original HAQ, HAQ with early termination (HAQ+E) and profiling based augmented DRL (P-ADRL) and the speedups of HAQ+E and P-ADRL over HAQ.

| | | Search | | | Finetune | | | Overall |
|---|---|---|---|---|---|---|---|---|
| | | episode | time (s) | speedup | epoch | time(s) | speedup | speedup |
| CifarNet | HAQ | 600 | 22464 | - | 1 | 13 | - | - |
| | HAQ+E | 103 | 3856 | 5.8X | 1 | 22 | 0.59X | 5.8X |
| | P-ADRL | 30 | 935 | 24X | 1 | 17 | 0.76X | 23.8X |
| ResNet20 | HAQ | 600 | 22680 | | 100 | 2900 | - | - |
| | HAQ+E | 68 | 2570 | 8.8X | 100 | 2900 | 1X | 8.8X |
| | P-ADRL | 13 | 342 | 66.4X | 5 | 140 | 20.7X | 64.1X |
| AlexNet | HAQ | 600 | 37800 | - | 43 | 19049 | - | - |
| | HAQ+E | 277 | 17451 | 2.16X | 100 | 44200 | 0.43X | 1.16X |
| | P-ADRL | 74 | 6147 | 6.15X | 35 | 16765 | 1.14X | 4.49X |
| ResNet50 | HAQ | 600 | 51900 | - | 100 | 302800 | - | - |
| | HAQ+E | 215 | 18597 | 2.79X | 100 | 312600 | 0.97X | 1.16X |
| | P-ADRL | 74 | 8580 | 6.04X | 8 | 23360 | 12.9X | 12.7X |

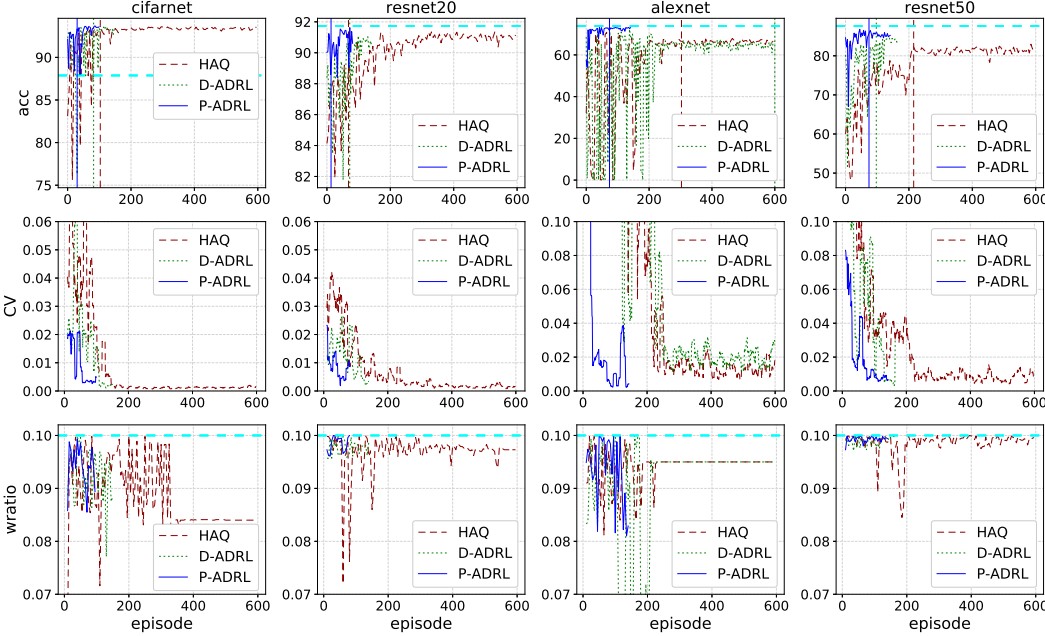

Figure 2: Results for search stage. Top Row: Accuracy. Middle Row: Coefficient of Variance (CV). Bottom Row: Weight Ratio (wratio) = Weight size under current configuration / Original weight size. The horizontal dashed light blue line is the accuracy of the original network before the quantization. The blue and green lines are the accuracy result for our strategies (P-ADRL and D-ADRL) while the dashed red line shows the result for HAQ. The two vertical lines mark the episodes at which the strategies terminated with early termination.

The accuracy given by P-ADRL is much stabler than that of the HAQ. We use the coefficient of variance (CV) to measure the stability of the inference accuracy. As shown in the second row of Figure 2, P-ADRL achieves a small CV much faster than HAQ on all four networks. This is the reason that P-ADRL terminates in fewer episodes than HAQ and HAQ+E do; this conclusion holds regardless of what variance threshold is used in the early termination as the CVs of P-ADRL are consistently smaller than those of HAQ.

The third row of Figure 2 shows the compression ratio after quantization at each episode for each of the three algorithms. The target ratio is set to 0.1. All three strategies fulfill the size limit requirement. By the end of the search stage, P-ADRL achieves a compression ratio similar to HAQ on large networks (ResNet20, AlexNet and ResNet50). On a small network CifarNet, the compression ratio given by P-ADRL is slightly larger than that of HAQ, but still smaller than 0.1.

**Finetuning Stage** After getting network precision from the search stage, we quantize the networks and finetune for 100 epochs. The learning rate is set as 0.001, which is the default setting from HAQ.

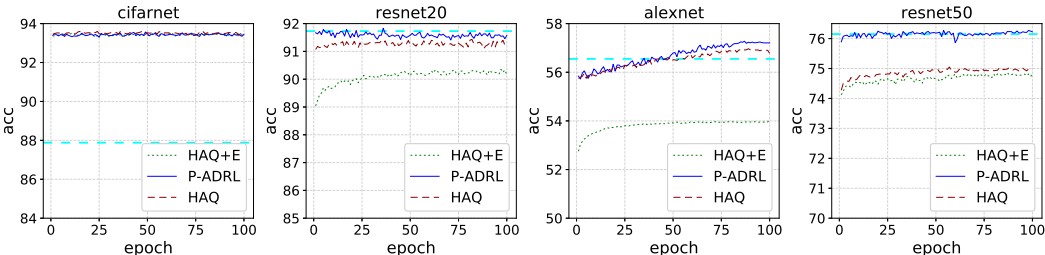

Figure 3: Finetune accuracy. (Horizontal dashed line: the original accuracy without quantization.)

Figure 3 illustrates the accuracy results of the finetune stages for three different precision settings: the one selected by P-ADRL; the one selected by HAQ+E and the one selected by the original HAQ. For CifarNet, all three achieve similar accuracies, higher than that of the original. For the other three networks, P-ADRL gets the largest accuracy, fully recovering the original accuracy. HAQ+E performs the worst, suffering at least 1% accuracy loss.

The sixth column of Table 2 gives the number of epochs each strategy needs for finetuning before reaching the accuracy of the original network; '100' means that the strategy cannot recover the original accuracy in the 100-epoch finetuning stage. P-ADRL takes 1–35 epochs to converge to the original accuracies, much fewer than HAQ and HAQ+E. Finetuning hence stops earlier.

## 5 CONCLUSION

This paper has demonstrated that simple augmentation goes a long way in boosting DRL for DNN mixed precision quantization. It is easy to apply ADRL to deeper models such as ResNet-101 and Resnet-152. In principle, it is also possible to extend the method to latest model architectures as the basic ideas of the reinforcement learning and the Q-value indicator are independent of the detailed architecture of the network model. Although the design of ADRL is motivated by DNN quantization, with appropriate augmentation schemes, the concept is potentially applicable to other RL applications—a direction worth future exploration.

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

# A APPENDIX

## A.1 DETAILS OF THE MIXED QUANTIZATION BITS FOR EACH MODEL

Figure 4 shows the comparison between the quantized networks selected using ADRL and HAQ. In each plot, the upper half shows the bit value being used for each layer for the network quantized with ADRL. The bottom half are the quantization configurations of the networks being quantized with HAQ.

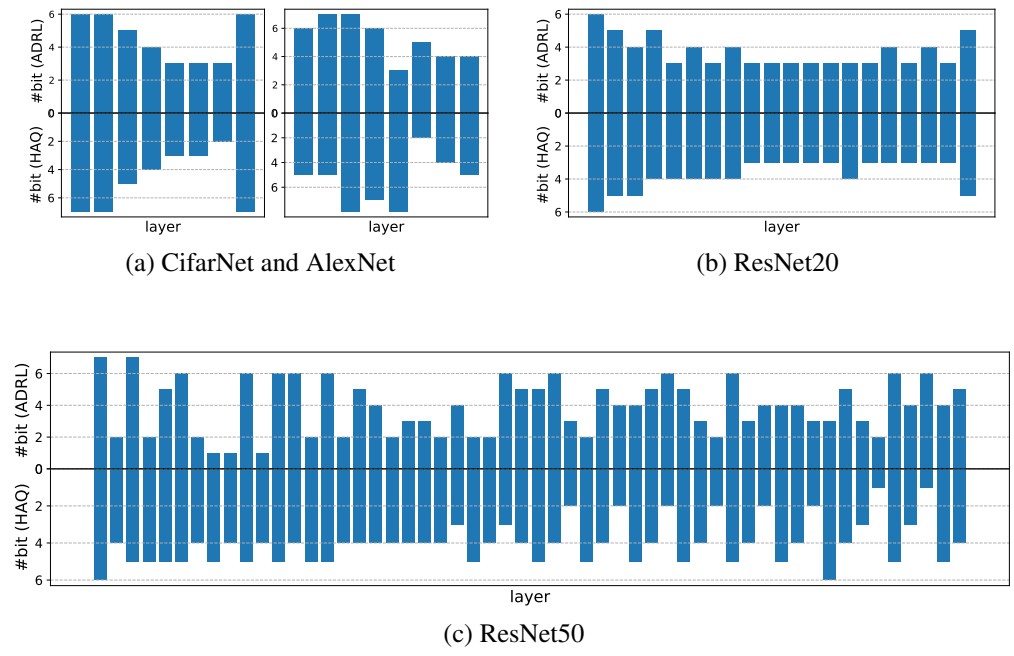

Figure 4: The bit value of each layer for (a) CifarNet (left) and AlexNet (right), (b) ResNet20 and (c) ResNet50. The methods for quantization are ADRL and HAQ.

## A.2 EVOLUTION OF THE SELECTED BIT VALUES

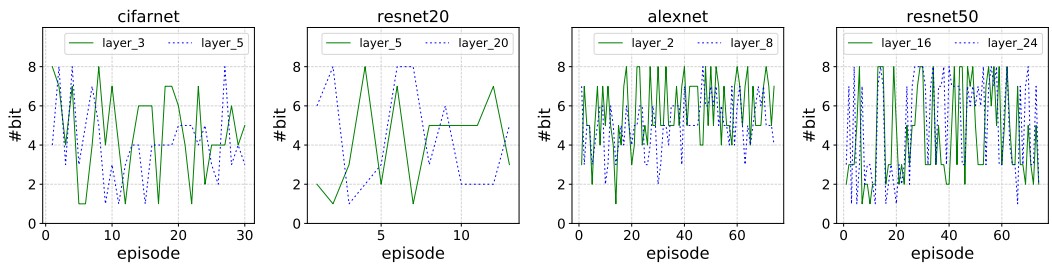

Figure 5: The bit values being selected in each search episode for the four networks.

Figure 5 illustrates how the selected bit values change during the search stage for some layers of the four networks. The bit value being selected for the same layer changes from one episode to the next. It spans well in the action search space.

