# OpenReview forum: "Simple Augmentation Goes a Long Way: ADRL for DNN Quantization"
_ICLR.cc/2021/Conference — ICLR 2021 Poster_

### Official Review · AnonReviewer1 · 2020-10-28
**effective approach, interested in further details on action proposals**

**Rating:** 7
**Confidence:** 3

**Review:**

This paper describes an improved way to determine weight quantization bit lengths using reinforcement learning, by injecting model evaluation directly into action selection.  Building upon a DRL setup where the action at each timestep corresponds to selecting a bit value for each layer, the method adds a "Q-value indicator" function Q~ that selects among a set of candidate actions, and filters based on the model performance quantizing the layer to each level.  This seems to form a hybrid between DRL and greedy search, using a greedy criteria Q~ to filter proposals made by the DRL agent.  Experiments show very good performance, with similar or better quantization levels and accuracy as other DRL-based methods and much faster runtime.

The method is well described overall, though I would have liked more details on the mu and Q networks including their initializations and how well the mu proposals span the action space initially (see questions below).

Likewise, I wonder if the mechanism of improvement is mostly through the initial guess provided by greedy search with Q~ at the beginning of training.  An alternative may be to edit the action network mu using a guess made by greedy profiling (see question 3 below) -- was anything like this explored?

Although I wonder whether the method may be able to be further simplified, I still find this a good paper overall, offering an effective way to reduce time to produce quantized models with no accuracy hit.  A few additional baselines of even simpler systems, including greedy search and greedy policy initialization, could help provide more context and assess the use of Q~ as a filter.



Additional questions:

1.  The descretized action space only has 8 values per layer.  What happens if one evaluates all 8 with the Q~ indicator and remove the candidate generation network?  Is this equivalent to a layerwise greedy search?  What is the final accuracy and quantization speed of this baseline?

2.  How is the last layer of the expanded actor function mu^ initialized?  Randomly, or in such a way that the initial outputs correspond to different discrete bit sizes?  How many candidate actions are there?  How well do the candidate values span the action space to allow for guidance by Q~, particularly in the beginning episodes?

3.  Instead of having multiple candidates, is it possible to introduce an output bias per layer in the actor function mu, initialized to the bit sizes determined by greedy profiler search with Q~ ?  That is, to use a function mu_l(s) = mu(s) + b(l), where b(l) = argmax_a Q~(a) for each layer l, then use regular DRL (not ADRL) with this mu_l that has an initial guess using greedy search from the profiler?

4.  I don't see where the system is encouraged to use fewer bits.  If more bits generally leads to higher accuracy, what makes the system learn to output smaller bits in its actions?  Both Q~ augmentation and the reward R = acc_quant - acc_orig depend only on accuracy.

5.  How different are the proposal actions from one another (do they span the space of possible bit lengths), and how do they change over the course of model selection?  Is the initial max according to Q~ used most of the time or are there times when the proposal index selected by Q~ changes from one episode to the next?

---

> ### Author Response · Authors · 2020-11-21
> **Response to Reviewer 1 [1/2]**
>
> We thank the reviewer for the positive feedback as well as the detailed comments and suggestions. We are happy to address them below:
>
> **1. The descretized action space only has 8 values per layer. What happens if one evaluates all 8 with the Q~ indicator and remove the candidate generation network? Is this equivalent to a layerwise greedy search? What is the final accuracy and quantization speed of this baseline?**
>
> It is important to notice what the candidate generation networks learn. It is not only the relation that the quantization has with the network accuracy, but also the relation with the size of the network. It is embodied in how the candidate generation networks are trained in the RL. In each episode, first, the Q-value indicator selects the candidate network that gives the highest accuracy for each layer. Note that after selecting the bit values for all the layers, the RL agent adjusts the bit values for each layer to generate a quantized model that **fulfills the compression ratio requirement**. It is the accuracy of that quantized model that is used to compute the reward and to train the candidate generation network. So by applying the Q~ indicator on only the candidates generated by the candidate generation networks, the design considers both accuracy and size.
>
> So if we remove the candidate generation networks from the design, the selected bit values would be always those that give the highest accuracy. It is hence hard to find a small enough model that fulfills the compression requirement; enumerating all possibilities would face an exponential search space (8^n where n is the number of layers).
>
> **2. How is the last layer of the expanded actor function mu^ initialized? Randomly, or in such a way that the initial outputs correspond to different discrete bit sizes? How many candidate actions are there? How well do the candidate values span the action space to allow for guidance by Q~, particularly in the beginning episodes?**
>
> The last layer of the expanded actor function mu^ is initialized randomly. In our experiments, we do observe that, with random initialization, the initial outputs tend to be different in the numbers of bits. (e.g. [1, 3, 8], [8, 7, 2], [3, 8, 2], [5, 3, 7], [1, 4, 2] for the first 5 layers of CifarNet.)
>
> In our experiments, we set the number of candidate actions to 3 (k=3) and demonstrate its efficacy.
>
> Based on our observations in the experiments, the candidate values span well in the action space. In our experiments, we have 3 candidates per layer. The 3 candidates tend to be different from each other. In the beginning episodes, the candidates for the same layers in two consecutive episodes tend to be different as well.
>
> **3. Instead of having multiple candidates, is it possible to introduce an output bias per layer in the actor function mu, initialized to the bit sizes determined by greedy profiler search with Q~ ? That is, to use a function mu_l(s) = mu(s) + b(l), where b(l) = argmax_a Q~(a) for each layer l, then use regular DRL (not ADRL) with this mu_l that has an initial guess using greedy search from the profiler?**
>
> If I understand correctly, here the ‘greedy profiler search with Q~’ is similar to the one mentioned in question 1, which evaluates all the 8 values. If this is the case, the b(l) will give the same value for the same layer in each episode and it tends to give larger values as more bits generally leads to higher accuracy. In this way, b(l) doesn’t help the system to learn anything new.
>
> If the number of values being evaluated by Q~ is less than 8, then how to decide which values being evaluated becomes another question. In this case, a multi-candidates actor function would be a better choice.
>
> Please correct me if I misunderstood the meaning of the suggested solution.
>
> **4. I don't see where the system is encouraged to use fewer bits. If more bits generally leads to higher accuracy, what makes the system learn to output smaller bits in its actions? Both Q~ augmentation and the reward R = acc_quant - acc_orig depend only on accuracy.**
>
> As explained in the answer to Q1, the system learns to generate bit values with the guidance of both the quantized model accuracy and the model compression ratio. Here is how it generates a network using fewer bits. After the agent selects the bit values for all the layers, it checks whether the quantized network fulfills the compression ratio requirement. If not, it will decrease the bit value layer by layer starting from the one that gives the least accuracy loss until the compression ratio of the quantized network is equal or smaller than the target ratio. Then it evaluates the network accuracy of the compressed network to compute the rewards. The final bit value of each layer forms the state used to train the agent. We added a paragraph in 'Implementation Details' in section 3.3 describing this process.

---

> > ### Author Response · Authors · 2020-11-21
> > **Response to Reviewer 1 [2/2]**
> >
> > **5. How different are the proposal actions from one another (do they span the space of possible bit lengths), and how do they change over the course of model selection? Is the initial max according to Q~ used most of the time or are there times when the proposal index selected by Q~ changes from one episode to the next?**
> >
> > As mentioned in the response to the Q2, the candidates tend to be different from each other within the same layer. And the candidates for the same layers in two consecutive episodes tend to be different as well. The candidates generated by the candidate generation network change from one episode to the next, so does the bit value selected by Q~ for each layer. We added a figure illustrating how the proposed actions evolve during the search stage in Appendix A2.

---

### Official Review · AnonReviewer3 · 2020-10-29
**Simple but effective method with strong performance improvement**

**Rating:** 6
**Confidence:** 3

**Review:**

========
Summary

This paper studies the DNN quantization using deep reinforcement learning. The paper proposes an augmented DRL which introduces a Q-value indicator to refine action selection. The proposed approach has been applied to several image classification baselines and has compared with several recent DRL based quantization approach, achieving a similar compression rate without accuracy decrease.  In addition, compared to previous methods, the learning speed has been improved by 4.5-64x.

=========
Pros

1. The paper is well written and easy to follow.  It is very clear even for an audience who may not be an expert on DNN quantization.
2. The idea is simple and reasonable to introduce an additional Q-value indicator to refine action selection.  Given the improvement in performance,  despite the simplicity, the method does provide great performance.


==========
Concerns/Comments

1.  Is there any result about the final quantization configuration of the compressed model? Is there any takeaway about the quantization pattern?

2.  What is the action space used in the experiments? How many bits were the model used?

3. In Table 1, could you provide the original accuracy of the models rather than the accuracy delta?  In addition, is that possible to quantize deeper models like ResNet-101 or ResNet 152? Is the method extendable to the latest model architectures such as ResNeXt, DenseNet, visual transformers?

---

> ### Author Response · Authors · 2020-11-21
> **Response to Reviewer 3**
>
> We thank the reviewer for the positive and valuable comments. We are happy to help addressing the concerns and comments as follows:
>
> **1. Is there any result about the final quantization configuration of the compressed model? Is there any takeaway about the quantization pattern?**
>
> The final quantization configurations are presented in Figure 4 in the Appendix. Looking at the final quantization configurations of both P-ADRL and HAQ in Figure 4, we can tell that the first layer and the last layer require more bits in general. For larger networks such as resNet50, we could have more aggressive compression (2 bits) for some inner layers.
>
> **2. What is the action space used in the experiments? How many bits were the model used?**
>
> In the experiment, each layer could be quantized into [1, 2, 3, 4, 5, 6, 7, 8] bits. The original model uses floating point numbers, which is 32 bits.
>
> **3. In Table 1, could you provide the original accuracy of the models rather than the accuracy delta? In addition, is that possible to quantize deeper models like ResNet-101 or ResNet 152? Is the method extendable to the latest model architectures such as ResNeXt, DenseNet, visual transformers?**
>
> We present the original accuracy of the models in the first row of Figure2 (illustrated with the horizontal dashed line).
>
> Yes, it is easy to apply ADRL to deeper models such as ResNet-101 and Resnet-152. In principle, it is also possible to extend the method to latest model architectures as the basic ideas of the reinforcement learning and the Q-value indicator are independent of the detailed architecture of the network model.

---

### Official Review · AnonReviewer4 · 2020-11-02
**It modifies a step in DRL based method that quantizes each networks with different bit numbers. A good engineering trick.**

**Rating:** 6
**Confidence:** 3

**Review:**

This work attempts to improve DRL based mixed-precision network compression. Compared to the standard action-critic method, he proposed method inserts an additional candidate evaluation step on k proposed actions as shown in Algorithm 1. As the paper title suggests, the proposed augmentation is “simple” yet practically brings non-trivial performance improvement.

I am not a real expert on RL. In general, I appreciate the idea in this work. It generates higher-quality action proposals by heuristically evaluating a few of them, either using a quick calculation of inference accuracy or direct quantization loss. The proposed idea is intuitive and well validated by experiments.

I have several suggestions for improving this work: the distance-based indicator is proved to be inferior in terms of performance, mentioned by authors. It is still suggested to report its accuracies in Tables 1 and 2, for a better view of two variants.  A few issues in experiments need clarification: is the reported time the total before convergence, or the per-episode time? The proposed method takes much fewer episodes to search but an additional q-value indicator brings more computations. More analysis regarding the complexity is required. There are a few ad hoc treatments in the method, such as memorizing the accuracy of specific bit value at a layer. Does it consider the evolution of the entire network (since the change of other layers have impact)? The theoretic analysis is based on assumptions that can be only empirically verified (yet not convincingly verified in the experimental sections). It is not clear to me the true value of section 3.2.

---

> ### Author Response · Authors · 2020-11-21
> **Response to Reviewer 4**
>
> We thank the reviewer for the helpful feedback and suggestions. We have incorporated the suggestions of the reviewer. We will address the comments and clarify the content of the paper as follows:
>
> **1. The distance-based indicator is proved to be inferior in terms of performance, mentioned by authors. It is still suggested to report its accuracies in Tables 1 and 2, for a better view of two variants.**
>
> Although we don’t report the accuracies with the distance-based indicator in Table 1 and 2, we do include the results (indicated by D-ADRL) from the search stage in Figure 2. The accuracy results presented in the first row of the figure show that D-ADRL can only lead to models with lower accuracy compared to the P-ADRL counterparts at the time of termination.
>
> **2. A few issues in experiments need clarification: is the reported time the total before convergence, or the per-episode time? The proposed method takes much fewer episodes to search but an additional q-value indicator brings more computations. More analysis regarding the complexity is required.**
>
> The reported time is the total time before convergence. It includes the time being spent on the q-value indicator. With the q-value indicator, the agent does need to run inference on a subset of test dataset for each candidate action for each layer per episode. However, the overhead becomes negligible compared to the time being saved by reduced search episodes. In addition, we introduced the memorization which further helped reduce the overhead of the q-value indicator. We added a discussion of the complexity into the ‘Implementation Details - Time complexity’ in section 3.3 in the revised version.
>
> **3. There are a few ad hoc treatments in the method, such as memorizing the accuracy of specific bit value at a layer. Does it consider the evolution of the entire network (since the change of other layers have impact)?**
>
> When using memorization, we don’t consider the evolution of the entire network. The stored accuracies are used for indicating layer-wise sensitivity on the bit values. The legitimacy of using such information is confirmed by the positive results in our experiments, as well as many previous studies (for example, [1-4]) on pruning and compression that use such results as indicators of layer-wise sensitivity.
>
> **Q4. The theoretic analysis is based on assumptions that can be only empirically verified (yet not convincingly verified in the experimental sections). It is not clear to me the true value of section 3.2.**
>
> Section 3.2 aims to give a theoretical explanation of the experimental results. We didn't present the reward results in the paper due to the space limit, but the Q-value indicator of ADRL does lead to higher reward compared to HAQ. Therefore, proposition 1 states that it would lead to smaller variance, which explains our observations shown in the second row of Figure 2. Proposition 2 gives an explanation of the faster convergence speed of ADRL compared to HAQ in the search stage (presented in Table 2).
>
> ***
> [1] Pruning filters for efficient convnets, Li, Hao and Kadav, Asim and Durdanovic, Igor and Samet, Hanan and Graf, Hans Peter, ICLR 2017.
> [2] Importance Estimation for Neural Network Pruning, Pavlo Molchanov, Arun Mallya, Stephen Tyree, Iuri Frosio, Jan Kautz, CVPR 2019.
> [3] Sensitivity-oriented layer-wise acceleration and compression for convolutional neural network, Wei Zhou, Yue Niu, Guanwen Zhang. IEEE Access 2019.
> [4] Hawq: Hessian aware quantization of neural networks with mixed-precision. Zhen Dong, Zhewei Yao, Amir Gholami, Michael W. Mahoney, and Kurt Keutzer. ICCV 2019.

---

### Author Response · Authors · 2020-11-21
**General response**

We thank all the reviewers for their positive feedback and insightful comments. We have uploaded a revision to the paper based on these comments and suggestions. We also replied to each reviewer below addressing the comments and questions.

Here we provide a summary of the major changes made to the paper:

1. We added a paragraph in ‘Implementation Details’ in section 3.3 describing how the system learns to use fewer bits.

2. We added a discussion of the time complexity of different approaches into the ‘Implementation Details - Time complexity’ in section 3.3.

3. We added a figure in Appendix A2 to show how the proposed actions evolve during the search stage for each of the four networks in our experiments.

---

### Decision · Program_Chairs · 2021-01-07
**Final Decision**

**Decision:**

Accept (Poster)

**Comment:**

This paper proposes a simple yet effective approach for determining weight quantization bit lengths using RL. All the reviewers agree that the simplicity and performance improvements are a strong plus point. There are some concerns on applicability which have been sufficiently handled by rebuttal. AC recommends accepting the paper.